# Sixty Years at the Rega Institute

**DOI:** 10.3390/v17020222

**Published:** 2025-02-05

**Authors:** Erik De Clercq

**Affiliations:** Rega Institute for Medical Research, KU Leuven, Herestraat 49, B-3000 Leuven, Belgium; erik.declercq@kuleuven.be

**Keywords:** interferon, poly(I).poly(C), suramin, ANPs, NRTTs, NtRTTs, NNRTTs, TDF, TAF, PrEP

## Abstract

I started my research career (in 1965) on interferon by identifying polyacrylic acid (PAA) as an interferon inducer. Poly(I).poly(C), discovered by Maurice Hilleman’s group, proved to be more potent as an interferon inducer, and through its mRNA, we were able to clone and express human β-interferon. The discovery of the reverse transcriptase (RT) by Temin and Baltimore (in 1970) brought me to the detection of suramin as a powerful RT inhibitor and enabled Sam Broder and his colleagues to identify suramin as the first inhibitor of HIV replication. In this capacity, it was subsequently superseded by AZT and other 2′,3′-dideoxynucleoside (ddN) analogs, including d4T. In collaboration with Antonín Holý, we discovered several acyclic nucleoside phosphonates as potent inhibitors of both HIV and HBV (hepatitis B virus) replication. In collaboration with Paul Janssen, we identified various non-nucleoside RT inhibitors (NNRTIs) of HIV-1 replication. Of the nucleotide RT inhibitors (NtRTTs), tenofovir emerged as the most promising congener. It was derivatized to its oral prodrugs TDF and TAF. To enhance their efficacy, they were combined with other anti-HIV drugs, and two of them were pursued (and found efficacious) in the Pre-Exposure Prophylaxis (PrEP) of HIV infections.

## 1. Introduction

While the university (KU Leuven) is celebrating its 600th anniversary in 2025, I have been at the Rega Institute for 60 years. It started in 1965, or actually in 1964, when Prof. De Somer invited me to the exam(ination) of microbiology (part II, virology) to join his research team.

## 2. Polyacrylic Acid

Interferon was discovered in 1957 by Isaacs and Lindenmann, with the influenza virus as the inducer [1]; apparently, it served as a regulator of its own replication [2]. It was recognized as an innate antiviral substance and incited the search for varying compounds of either viral or (preferably) nonviral origin that could induce the same antiviral principle. In this perspective, Merigan and Regelson described the synthetic polyanion, pyran copolymer, as an inducer of interferon in mice and men [3,4]. In August 1966, I started my search for interferon inducers in the lab of Prof. Piet De Somer, and this search promptly led to the identification of polyacrylic acid (PAA) (Figure 1) as an interferon inducer (in mice). This finding encouraged me to continue my research work in the lab of Prof. De Somer, and not to further pursue, albeit partially or temporarily, a clinical career in Internal Medicine with Prof. Josué Van den Broucke. In 1967, Piet De Somer had met Thomas C. Merigan at a meeting in Fort Lauderdale, and they must have discussed the possibility of me staying at Stanford University (Palo Alto) to get further experience with inducers of interferon. On 3 September 1968, we, Lili and I, flew to Palo Alto, 2 days after our marriage on 31 August 1968. Still in 1968, we [5,6] published that PAA, like pyran copolymer, behaved as a potent interferon inducer in mice. I demonstrated that both PAA and mouse interferon suppressed the formation of pox tail lesions in mice infected with the vaccinia virus [7].

## 3. Poly(I).poly(C)

In 1967, before I left for Stanford University, I was highly impressed by the observations of Maurice Hilleman’s group that double-stranded RNAs of varying origin behaved as potent interferon inducers, the “primus inter pares” being poly(I).poly(C) (Figure 2) [8]. Of the two components, poly(I) was clearly more important than poly(C) [9]. Interferon induced by the double-stranded RNA was also found effective against murine tumors induced by the Moloney sarcoma virus [10]. Although highly potent as an interferon inducer, poly(I).poly(C) was never exploited or commercialized for medical use. Various attempts (by chemical modification) to dissociate the interferon-inducing capacity of poly(I).poly(C) from its toxicity invariably failed (although a lesser toxic derivative was temporarily pursued as Ampligen^®^), so that the double-stranded RNA approach as a source for an effective (and safe) medicine eventually vanished.

While poly(I).poly(C) did not fulfill its original promise as an antiviral drug, it proved extremely useful as an in vitro inducer of interferon mRNA, thus leading to the cloning of human interferon β [11] and its expression [12]. As a side product in the interferon β induction process, interferon β2 was identified [13]. Interferon β2 would be better known as interleukin-6, whereas the human interferon β would be medically used in the treatment of multiple sclerosis (MS).

## 4. Suramin

In 1970, a few months before I was scheduled to return from Stanford to the Rega Institute, I came across two papers in Nature [14,15], revealing the existence of an enzyme, named reverse transcriptase (RT), catalyzing the RNA-dependent DNA synthesis (in 1975, Temin and Baltimore would receive the Nobel Prize for Physiology and Medicine for this discovery, together with Renato Dulbecco). The discovery of RT was then confirmed in different laboratories, including our own. Sol Spiegelman’s group found the enzyme in all cancers they looked at and concluded that the RT was responsible for these cancers.

Prominent among several compounds I had identified as potent RT inhibitors was suramin (moranyl) (Figure 3); guided by Spiegelman’s hypothesis that the RT accounted for the origin of cancer, I injected murine leukemia cells into mice and evaluated whether suramin influenced the death rate of these mice. To my disappointment, it did not. When Bob Gallo visited my lab several years later (in 1977), I told him about my experience(s) with suramin. He thought that I should (attempt to) publish the RT results in the journal “Cancer Letters”, for which he was then the Editor, and so I finally did, 4 years after I had generated the results [16]. Disappointed by the negative results I had obtained with suramin in the treatment of leukemia in mice, I had lost my interest in suramin, until, in 1984, I got a phone call from a certain Sam Broder (National Cancer Institute, NIH) to congratulate me with the discovery of suramin as an RT inhibitor. While originally enigmatic, the reason for this phone call became quickly clear. The journal Science had just accepted for publication an article [17], in which Sam Broder (R.C. Gallo being co-author) had reported that, inspired by my RT paper on suramin [16], they had evaluated (and found suramin active) against HIV (then called HTLV-III) in vitro in cell culture. One year later, Sam Broder also reported that suramin was effective in vivo in suppressing HTLV-III infection [18].

## 5. AZT

Thereupon, the attention of Sam Broder and colleagues shifted from suramin to azidothymidine (AZT) (Figure 4) as a potent (and less toxic) inhibitor of HTLV-III replication. Although already announced in June 1985, the paper was published in October 1985 [19]. Azidothymidine was not unknown to me, as I had published that the compound, like several other nucleoside analogs, had no antiviral activity against herpes simplex virus (HSV) or vaccinia virus (VV) [20]. If I had tested the compound for in vivo activity against the Moloney sarcoma virus (MSV) for which I had developed the strategy when still being at Stanford [21], I could have found its anti-retroviral efficacy. With the 5′-triphosphate of AZT, its inhibitory effect on the HIV-1 RT was ascertained. This work was accomplished in 1987 [22].

Following AZT, various other 2′,3′-dideoxynucleosides (ddNs), i.e., ddI, ddA, ddG, ddT, and ddC, were found to inhibit HTLV-III replication [23]. Promptly after the PNAS October 1985 issue [19] had appeared, P. Herdewijn at the Rega Institute had synthesized d4T (2′,3′-didehydro-2′,3′-dideoxy thymidine) (Figure 5). This compound was sent to Jan Balzarini, who then worked in Sam Broder’s laboratory. The results obtained with d4T against HTLV-III using ATH-8 cells were not overwhelming, patentable, or publishable. This all happened within the first months of 1986. Then, Masanori Baba arrived in the lab in August 1986 to start his fellowship studies. He should have started earlier, but his stay was delayed because his son (Chung) had developed Kawasaki’s syndrome. When he arrived, I asked him to recheck the anti-HIV activity of d4T in the MT-4 cells we had just obtained from Dr. Luc Montagnier, which were sent to him by Prof. Naoki Yamamoto (Tokyo).

To our surprise, d4T demonstrated a much higher potency and selectivity in MT-4 cells than obtained in the ATH-8 cells, and this time, I found the results publishable and patentable. In a few weeks, Masanori had the results written up in a paper that we then submitted for publication in Biochemical and Biophysical Research Communications (BBRC) to the Editor Dr. A. Sols (Madrid). Simultaneously, I sent the paper to our patent attorney in The Hague, Mr. C.W. Bruin (Arnold & Siedsma), with the request to file our findings for a patent application. This all happened on 25 November 1986. The paper was accepted for publication in BBRC by Dr. A. Sols, and it was published in the journal on 15 January 1987, but Mr. Bruin did not comply with our request on 25 November 1986 to file a patent application. He finally did so in January 1987 after the BBRC paper had been published [24]. That we were the first to report the anti-HIV activity of d4T (stavudine) was clearly evident from the fact that Lin, Schinazi, and Prusoff [25] and Hamamoto et al. [26] published their findings 6 months later than Baba et al. [24]. Nevertheless, Bristol-Myers (before it had become Bristol-Meyers Squibb (BMS)) decided to commercialize d4T for the treatment of HIV infections in collaboration with Yale University.

## 6. BVDU

In May 1976, I was invited to attend the Symposium on Synthetic Nucleosides, Nucleotides, and Polynucleotides in Göttingen. I was the only medical doctor at this meeting, which was attended by nucleoside/nucleotide chemists from the West and the East and where I met for the first time Dr. Richard (“Dick”) Walker (Birmingham, UK) and Dr. Antonín Holý (IOCB, Prague). In retrospect, the meeting in Göttingen was the most successful event of my scientific life, not because of the discovery of any of the chemical compounds that would ultimately prove effective as antivirals, but because of the discovery of the chemists who synthesized them. One of these compounds [BVDU, (*E*)-5-(2-bromovinyl 2′-deoxyuridine)] (Figure 6) originated in Birmingham (UK), where it was synthesized by Phil Barr, A.S. Jones and R.T. Walker. It proved highly active in inhibiting the replication of herpes simplex virus (HSV), both in vitro and in vivo [27]. These findings were published shortly (1–1.5 years) after Elion and Schaeffer had reported the selective activity of acyclovir against HSV [28,29]. As compared to acyclovir, BVDU proved roughly equally active against HSV-1 but much less so against HSV-2, thus minimizing its potential efficacy in the treatment of genital herpes. For this reason, Searle (US) would finally abandon the further clinical development of BVDU for the treatment of HSV infections. What they ignored, however, was that BVDU had marked potential for the treatment of varicella-zoster virus (VZV) infections, although anecdotal evidence had pointed to this potential efficacy [30]. This was later confirmed in a clinical trial where intravenous acyclovir was assessed in comparison with oral BVDU [31].

That Searle US was not interested in further developing BVDU as an anti-HSV drug was communicated at Searle UK in High Wycombe by Dan Azarnoff (then CEO of Searle US) at a breakfast meeting that both Prof. P. De Somer and I attended, some day in April 1984. To alleviate the disappointment, Searle handed over to us the stock of BVDU that they had synthesized in the meantime. With Searle having ended the agreement, I started—in vain—to get other pharmaceutical companies interested in BVDU, until I received the news from the DDR (Deutsche Demokratische Republik) that in the meantime they had marketed BVDU (which they named brivudine) under the tradename of Helpin^®^ for the treatment of VZV infections (herpes zoster) in immunocompromised patients. The company responsible for this marketing was Berlin Chemie (DDR) and they originally believed the formula of their BVDU was 5-(1-bromovinyl 2′-deoxyuridine). This erroneous identification did not prevent the marketing of Helpin^®^ in the DDR, but with the fall of the Berlin Wall in 1989 and the take-over of Berlin Chemie by the Italian company Menarini, as well as the worldwide marketing of brivudine (BVDU) as an oral drug for the treatment of VZV infections (herpes zoster) in both immunocompetent and immunosuppressed patients, the BVDU patent rights had to be assigned to their legal owners, being Birmingham University and the Rega Institute. BVDU is still the most potent among the anti-VZV drugs that are clinically used and is now available as a generic drug in most countries worldwide (except for the US and UK).

## 7. DHPA, (*S*)-HPMPA

At the meeting in Göttingen, I also met (for the first time) Dr. Antonín Holý from Czechoslovakia, and we agreed that I would evaluate for antiviral activity the compounds that would be made available by Dr. Holý. After a few months, Tony Holý forwarded three compounds, one of which [(*S*)-2′,3′-dihydroxypropyladenine (DHPA)] (Figure 7) proved antivirally active, particularly against VV and VSV (vesicular stomatitis virus). It would later be elucidated that the antiviral activity was due to an inhibition of the S-adenosylhomocysteine hydrolase, resulting in an inhibitory effect on the (viral) mRNA, thus causing an inhibition of virus replication. Our finding was promptly patented and published in Science [32]. Thereupon, Holý persuaded a local pharmaceutical company, Lachema, to commercialize DHPA, as Duviragel^®^, for the topical treatment of herpes labialis (cold sores), although we never gathered experimental evidence for any inhibitory efficacy of the compound in the treatment of herpes labialis. Duviragel^®^ was abandoned after the dissociation of Czechoslovakia in the Czech Republic (Czechia) and the Slovak Republic (Slovakia).

Yet, the collaboration between A. Holý and myself continued and would, in 1986, lead to the discovery of the acyclic nucleoside phosphonates (ANPs) as potent inhibitors of DNA virus replication. The prototype of the ANPs was (*S*)-9(3-hydroxyl-2-phosphonyl methoxy propyl)adenine [(*S*)-HPMPA] (Figure 8), which was reported in Nature [33]. Professor Piet De Somer would never witness the publication of this paper (in October 1986), as he died the year before (on 13 June 1985). (*S*)-HPMPA was never approved for medical use, although anecdotal experience had pointed to its clinical efficacy in the treatment (upon topical application) of adenovirus keratoconjunctivitis.

## 8. (*S*)-HPMPC

One of the reasons for not selecting (*S*)-HPMPA for clinical evaluation (and development) was that several other ANPs had come along [34], the cytosine counterpart, (*S*)-HPMPC, being equally promising but considered less toxic (in mice) than (*S*)-HPMPA. (*S*)-HPMPC (Figure 9) looked quite attractive as a potential inhibitor of the replication of human cytomegalovirus (HCMV) [35], and in 1996 it was finally approved for the treatment of HCMV retinitis in AIDS patients [36]. (*S*)-HPMPC, in the meantime called cidofovir and marketed as Vistide^®^, has become available “off-label” for the treatment of other DNA virus infections, i.e., poxviruses and papillomaviruses. As to the poxviruses, cidofovir was from the start reserved as a possible treatment option should smallpox (variola virus) re-emerge, and J. Neyts and E. De Clercq found cidofovir therapeutically efficacious in the treatment of the complications of VV infections in immunosuppressed mice [37]. Stittelaar et al. also found cidofovir superior to vaccination (with VV) in inhibiting mousepox virus infection in monkeys [38].

After Van Cutsem et al. had demonstrated a dramatic regression of the hypopharyngeal and esophageal papilloma lesions in a patient with human papillomavirus (HPV) infection [39], topical application of cidofovir was successfully used in the treatment of laryngeal HPV infections [40], as well as genital, i.e., cervical HPV infections (i.e., cervical intracellular neoplasia (CIN) stage III) [41].

Because of its limited oral bioavailability, cidofovir has to be administered by the parenteral (i.e., intravenous) or topical route. In its prodrug form, brincidofovir (CMX001) can be administered perorally, and it has been approved for the (oral) therapy of monkeypox (Mpox) virus infections [42].

## 9. PMEA, (*R*)-PMPA

Simultaneously with (*S*)-HPMPA, 9-(2-phosphonylmethoxyethyl)adenine (PMEA) (Figure 10) was published as an antiviral drug, particularly for its activity against retroviruses [33]. The anti-retroviral activity was further documented by Pauwels et al. [43], and its mechanism of action was established by Balzarini et al. [44]. PMEA (adefovir) was converted to its dipivoxil derivative to enhance its oral bioavailability, and the resulting adefovir dipivoxil was submitted (by Gilead Sciences) to the US Food and Drug Administration (FDA) for approval to be marketed in the therapy of HIV infections. Eventually it failed to be approved, essentially because of two reasons: (i) its dosage required to inhibit HIV replication (60 mg daily) was considered too high and hence too toxic, and (ii) Gilead Sciences had in the meantime developed a more potent and less toxic ANP, namely (*R*)-PMPA [(*R*)-9-(2-phosphonylmethoxypropyl)adenine, tenofovir], which they had derivatized to its oral prodrug, tenofovir disoproxil fumarate (TDF) that could be safely administered at a higher dosage (300 mg daily). (*R*)-PMPA (Figure 11) had first been reported for its anti-HIV activity by Balzarini et al. [45], and its oral prodrug, the disoproxil derivative, had been described by Naesens et al. [46] and Robbins et al. [47]. TDF was approved by the US FDA in 2001 and commercialized by Gilead as Viread^®^. In the meantime, adefovir dipivoxil was found to inhibit hepatitis B virus (HBV) at a much lower dosage (10 mg daily) than originally planned for the treatment of HIV infections. It was then approved in 2002 and marketed for the treatment of (chronic) HBV infections under the trade name Hepsera^®^ (adefovir dipivoxil).

Following TDF, a second tenofovir prodrug, tenofovir alafenamide (TAF, Vemlidy^®^) was commercialized by Gilead for the treatment of HIV and HBV infections. As compared to TDF, TAF was much more effective [48], so that it could be dosed at 25 mg daily. The reason for its unique behavior as an anti-HIV and anti-HBV drug was that it was specifically metabolized in both lymphocytes and hepatocytes via tenofovir to the active metabolite, tenofovir diphosphate, thus achieving its inhibitory effect on the replication of HIV and HBV, respectively [49].

## 10. TIBO and HEPT Derivatives

In the fall of 1986 (5 November 1986), I gave a lecture at Janssen Pharmaceutica in Beerse (Belgium), and I had discussions with Dr. Paul Janssen, over lunch and dinner, when he expressed his desire to collaborate on the development of the “ideal” drug for the treatment of HIV infections.

It took 2 years before all the modalities of a collaboration between the Janssen Research Foundation and the Rega Institute were formalized, but with access to the chemical library of Janssen Pharmaceutica, we were able to identify TIBO (tetrahydroimidazobenzodiazepinone) (Figure 12) by 1 February 1990 as a novel inhibitor of HIV replication [50]. The compound appeared to act as an RT inhibitor that was specifically active against HIV-1 (and thus not HIV-2). That it specifically acted as an RT inhibitor was further ascertained by Debyser et al. [51].

Although clinically effective against HIV-1 replication, the TIBO derivatives were not further pursued for clinical development, primarily because their chemical synthesis was considered too cumbersome. Instead, novel HIV-1 RT inhibitors, α-anilinophenylacetamide (α-APA) derivatives, were reported [52].

Independently from the TIBO derivatives, we had described by the end of the 1980s a new class of HIV inhibitors for which we originally did not have any *modus operandi*, the HEPT derivatives [1-[2-hydroxyethoxy)methyl]-6-(phenylthio)thymine] (Figure 13) [53,54]. In the early 1990s, it became increasingly evident that the HEPT analogs reacted akin to the TIBO derivatives, as specific inhibitors of the HIV-1 RT [55,56]. I had noticed some similarity in the conformation of the HEPT and TIBO derivatives, which when I discussed this similarity with Dr. Paul Janssen, he labeled this interpretation as “pure phantasy” from my side. Yet, I could not resist attempting to point to overlapping conformations of the TIBO and HEPT derivatives [57]. Eddy Arnold and his coworkers [58] provided in 1995 a more structural basis for the concept that all the non-nucleoside RT inhibitors (NNRTIs) would adhere to a similar, “butterfly-like” conformation.

Chemical modifications of the HEPT analogs finally led to emivirine (MKC-442) [59] that progressed to phase III clinical trials for the therapy of HIV-1 infections, moving from Mitsubishi Kasei Corporation (MKC) to Wellcome, Triangle Pharmaceuticals, and eventually Gilead Sciences, where its further clinical development was halted, simply because it was no longer competitive as a potential anti-HIV drug. Within the Janssen Pharmaceutica company (Johnson & Johnson), TIBO followed an equally meandrous pathway [60], eventually coming close to fulfilling Dr. Paul Janssen’s ultimate dream, that is the ideal compound for the treatment of HIV infection (rilpivirine). With tenofovir (the nucleotide RT inhibitor, available as TDF and TAF) on the one hand and the NNRT inhibitor rilpivirine on the other hand, it appeared highly desirable to combine both drugs in the treatment of HIV infections [49].

## 11. Drug Combinations in the Therapy of HIV Infections

Gilead Sciences started to market its first combination, consisting of TDF with (-)FTC (emtricitabine, Emtriva^®^) for the therapy of HIV infections, namely Truvada^®^ in 2004. It was followed in 2006 with the combination of TDF, emtricitabine, and efavirenz, called Atripla^®^. Then followed in 2010 the combination of TDF, emtricitabine, and rilpivirine, called Complera^®^ in the US and Eviplera^®^ in the EU. The quadruple combination of TDF, emtricitabine, etravirine, and cobicistat (named Stribild^®^) followed in 2012. As soon as TAF had become available, it was substituted for TDF in its combinations with emtricitabine, rilpivirine, etravirine, cobicistat and darunavir, thus leading to the marketed products Descovy^®^ (TAF plus emtricitabine), Odefsey^®^ (TAF plus emtricitabine plus rilpivirine), Genvoya^®^ (TAF plus emtricitabine plus etravirine plus cobicistat), and Symtuza^®^ (TAF plus emtricitabine plus etravirine plus darunavir). Other drug combinations have recently been marketed for the therapy of HIV infections [61]. The rationale for using drug combinations in the therapy of HIV infections is similar to that for underlying the combined use of antitubercular drugs in the treatment of *Mycobacterium tuberculosis* infections: (i) to obtain synergistic activity between different drugs acting at different targets by different modes of action; (ii) to lower the dosages of the individual compounds, and hence, mitigate their toxic side effects; and (iii) to diminish the likelihood of drug resistance emergence.

## 12. Chemoprophylaxis of HIV Infections

Two drug combination formulations are currently available for the prevention of HIV infection: Truvada^®^ (TDF plus emtricitabine) and Descovy^®^ (TAF plus emtricitabine). Their prophylactic use is often referred to as PrEP (Pre-Exposure Prophylaxis). PrEP should be advised in all individuals who are at (increased) risk for being infected with HIV. That tenofovir might be effective in the prevention of HIV infections could be predicted from the observations of Tsai et al. [62], that (*R*)-PMPA (later named tenofovir) was able to reduce simian immunodeficiency virus (SIV) infection from 100% to 0% in *macacus rhesus* monkeys if administered subcutaneously shortly (within 1–2 days) before or after virus inoculation. This article was published on 17 November 1995. Seventeen years later, on 16 July 2012, the US FDA approved the use of tenofovir (in combination with emtricitabine) for the prophylaxis of HIV infections (in humans).

## 13. Conclusions

With the perspectives for the development of an effective vaccine against HIV waning rather than strengthening, the potential of PrEP with tenofovir plus emtricitabine to prevent HIV infection is acquiring increased attention. A long-lasting prevention of HIV infection could also be envisaged with biannual subcutaneous injections of lenacapavir. Although biannual injection of lenacapavir was reported not to lead to resistance after one year [63], resistance to lenacapavir was noted even after only 3 weeks on lenacapavir in another study [64]. With the lenacapavir resistance problem unsettled, it may be advisable to combine subcutaneous lenacapavir injections with peroral administration of TDF (or TAF) plus emtricitabine as a long-lasting approach to prevent HIV infection.

## Figures and Tables

**Figure 1 viruses-17-00222-f001:**
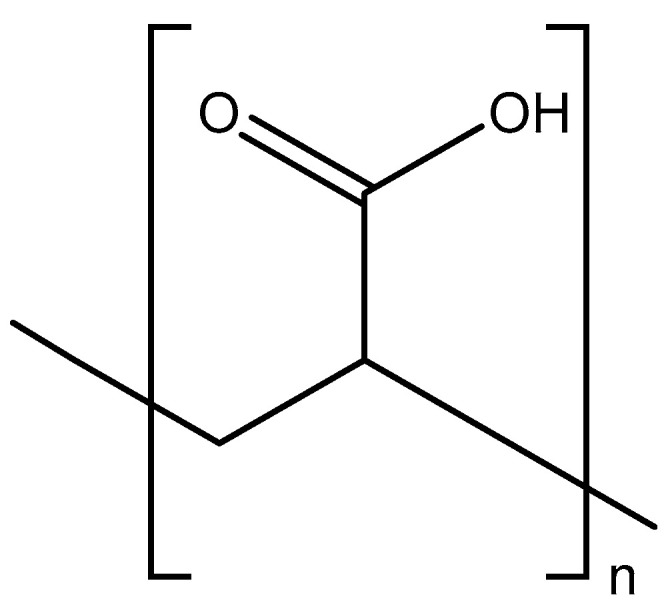
Polyacrylic acid (PAA).

**Figure 2 viruses-17-00222-f002:**
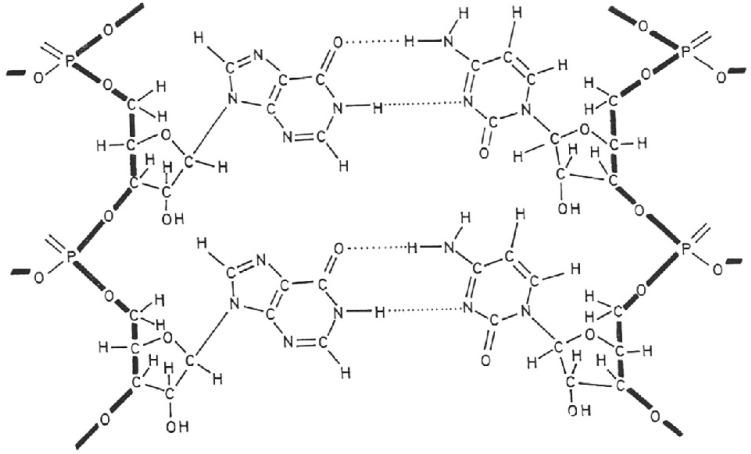
Poly(I).poly(C).

**Figure 3 viruses-17-00222-f003:**
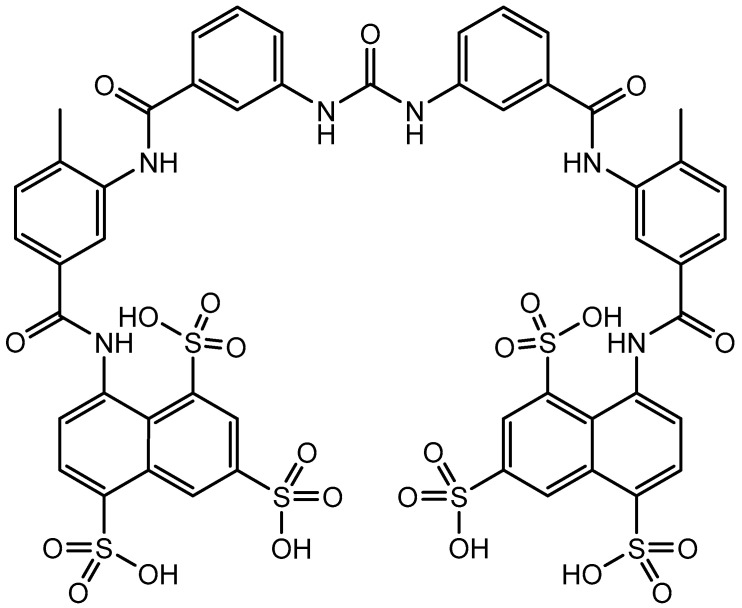
Suramin.

**Figure 4 viruses-17-00222-f004:**
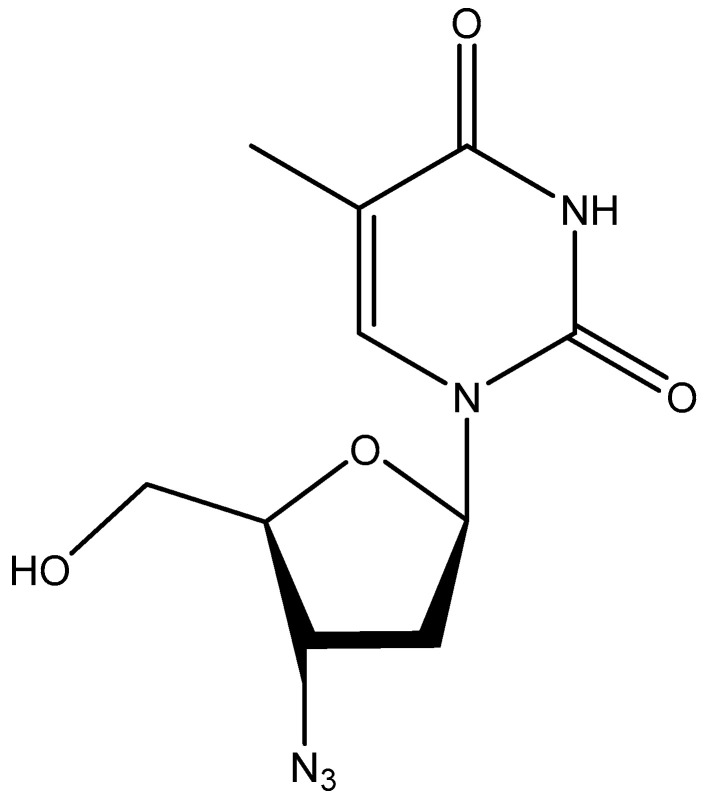
Azidothymidine (AZT).

**Figure 5 viruses-17-00222-f005:**
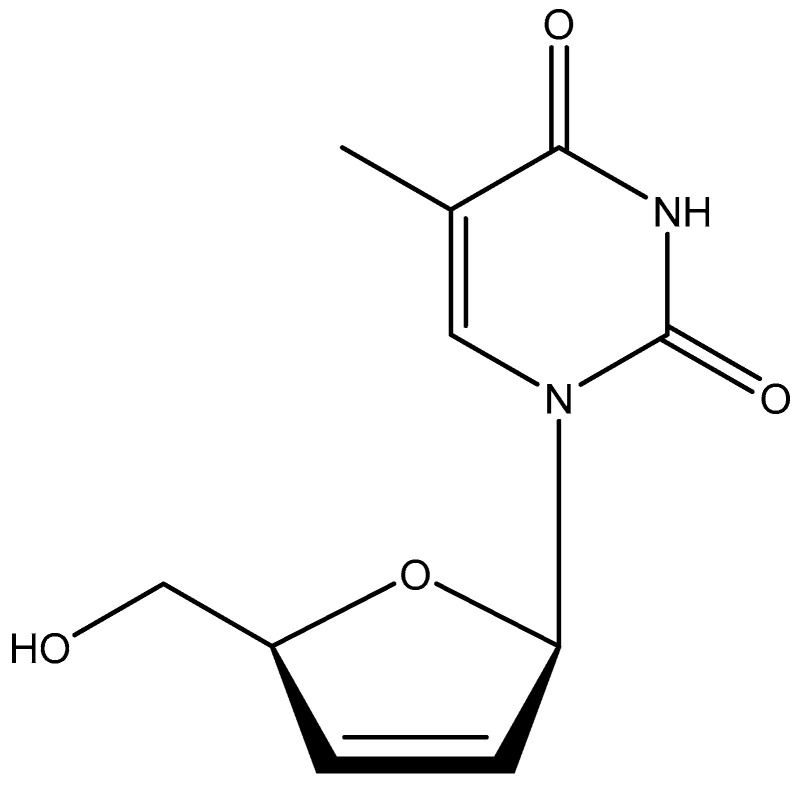
D4T (2′,3′-didehydro-2′,3′-dideoxy thymidine).

**Figure 6 viruses-17-00222-f006:**
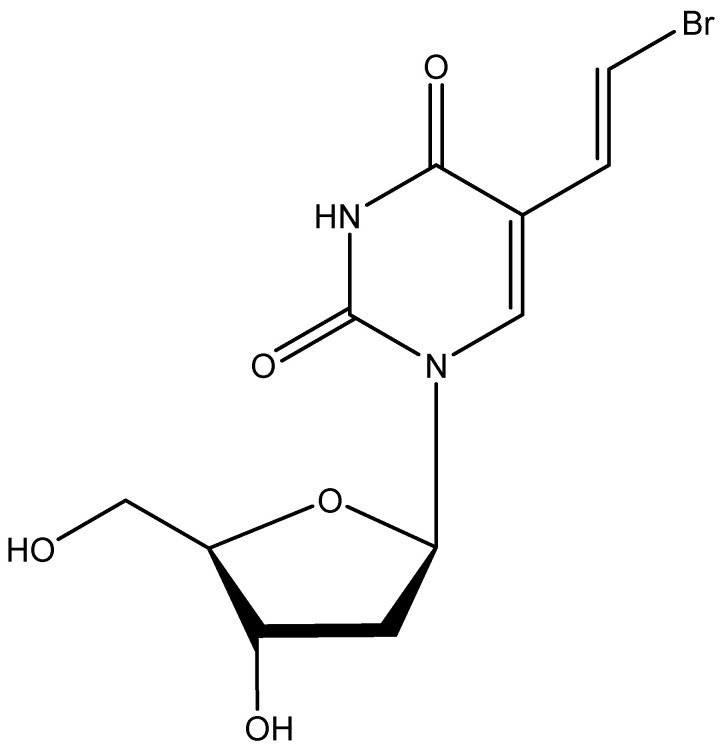
BVDU [(*E*)-5-(2-bromovinyl 2′-deoxyuridine)].

**Figure 7 viruses-17-00222-f007:**
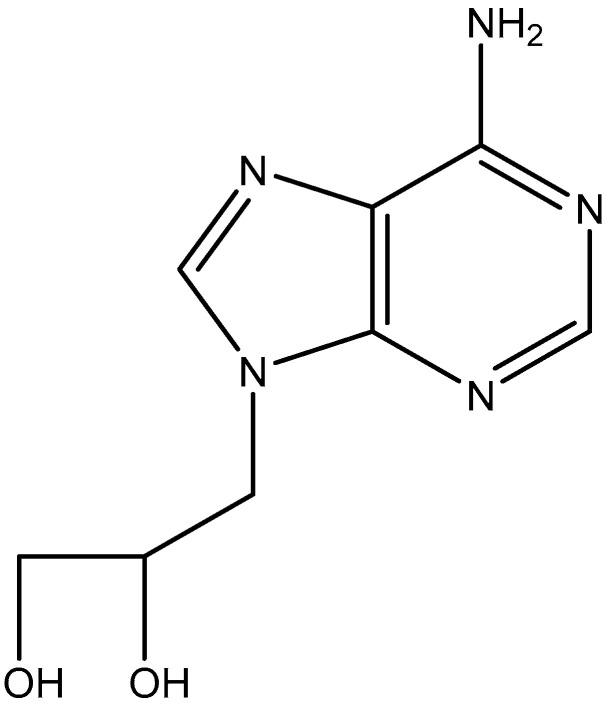
DHPA [(*S*)-2′,3′-dihydroxypropyladenine].

**Figure 8 viruses-17-00222-f008:**
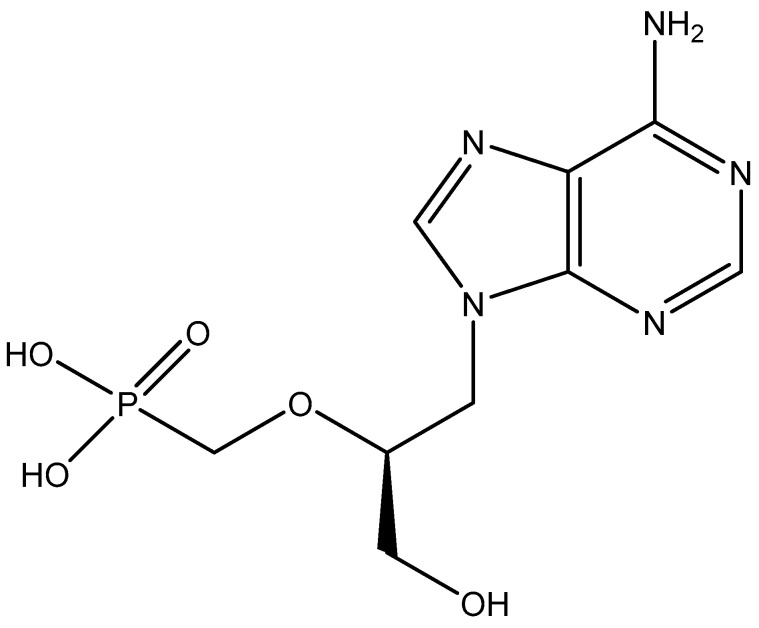
(*S*)-HPMPA [(*S*)-9(3-hydroxyl-2-phosphonyl methoxy propyl)adenine].

**Figure 9 viruses-17-00222-f009:**
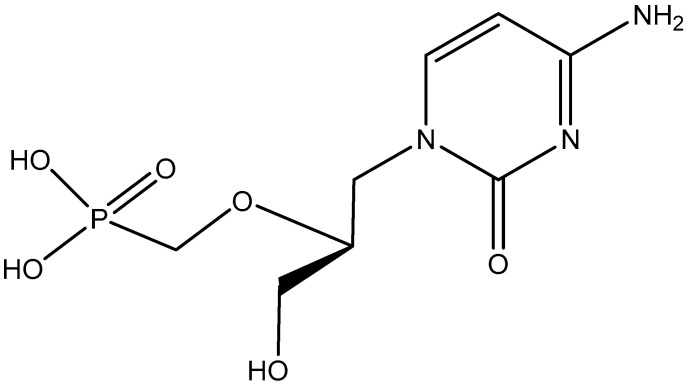
(*S*)-HPMPC (cidofovir).

**Figure 10 viruses-17-00222-f010:**
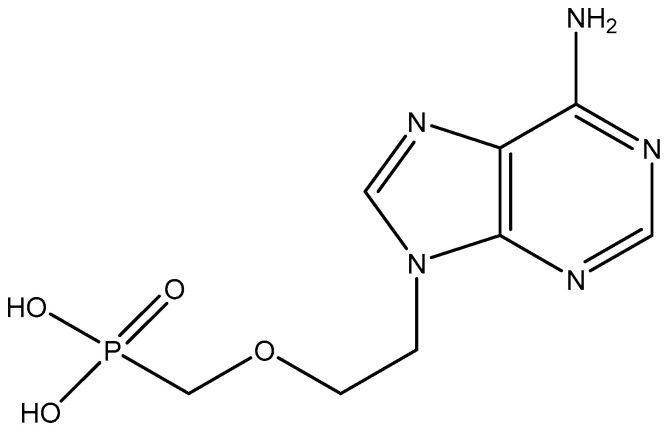
PMEA [9-(2-phosphonylmethoxyethyl)adenine].

**Figure 11 viruses-17-00222-f011:**
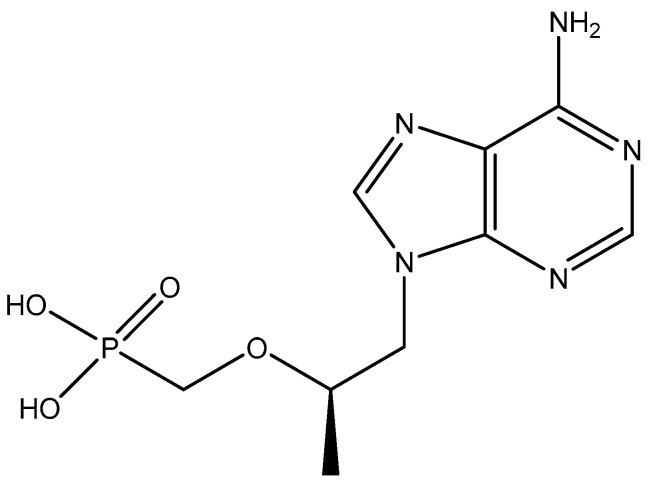
(*R*)-PMPA [(*R*)-9-(2-phosphonylmethoxypropyl)adenine].

**Figure 12 viruses-17-00222-f012:**
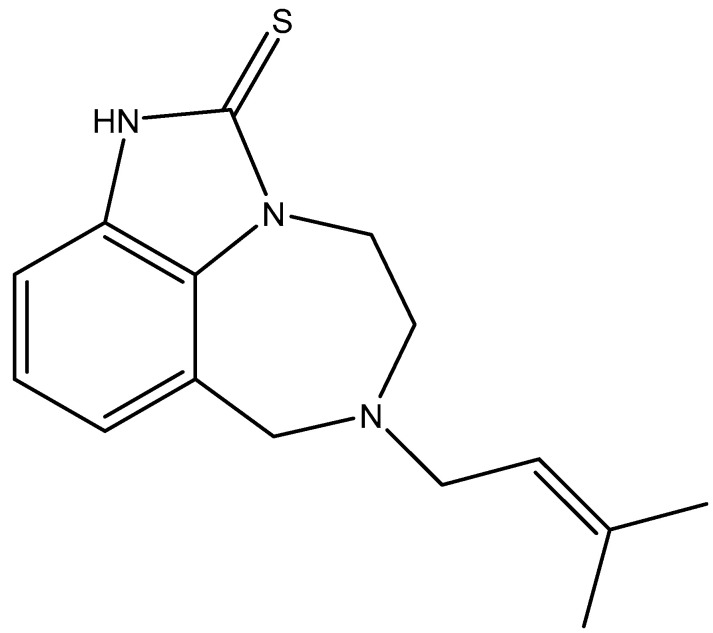
TIBO (tetrahydroimidazobenzodiazepinone).

**Figure 13 viruses-17-00222-f013:**
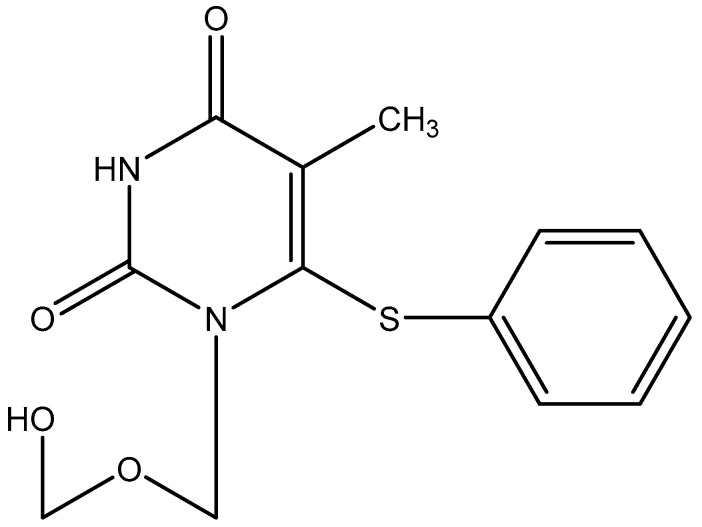
HEPT [1-[2-hydroxyethoxy)methyl]-6-(phenylthio)thymine].

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
