# Peer review of "Sixty Years at the Rega Institute"

_viruses, 2025, doi:10.3390/v17020222_

Round 1
Reviewer 1 Report
Comments and Suggestions for Authors
pg 11, Combinations for Therapy of HIV Infection.
Author should mention the ground breaking success of the Merck 035 trial and the Aids Clinical Trials Group -USA(ACTG) trial for HIV infection showing that the combination of a protease inhibitor, Indinavir, and two nucleoside RT inhibitors, ZDV and 3TC, could effectively inhibit HIV replication and lead to long term remission in HIV infected patients.
Gulick, Mellors, Havilar ,Eron, Emini and Chadakewitz. Potent and sustained antiretroviral activity of indinavir(IDV) in combination with zidovudine(ZDV) and lamivudine(3TC). Third conference on Retroviruses and Opportunistic Infections, Washington,D.C. 1,1996: 162
Hammer, Squres,Hughes et al, A controlled trial of two nucleoside analogues and indinavir in persons with HIV infection and CD4 cells of 200 per cubic millimeter or less. NEJM, 1997, 337, pg 725-733
Author Response
Comment 1:
pg 11, Combinations for Therapy of HIV Infection.
Author should mention the ground breaking success of the Merck 035 trial and the Aids Clinical Trials Group -USA(ACTG) trial for HIV infection showing that the combination of a protease inhibitor, Indinavir, and two nucleoside RT inhibitors, ZDV and 3TC, could effectively inhibit HIV replication and lead to long term remission in HIV infected patients.
Response 1:
I would like to thank the Reviewer for their suggestion. I have asked the Editor to add the suggested references as an addendum to the publication.
Reviewer 2 Report
Comments and Suggestions for Authors
This article is authored by a man who has long been regarded as a living legend among scientists dedicated to the study and treatment of HIV infection. The creator of one of the most effective antiretroviral drugs, tenofovir, he recounts the history of developing HIV treatments—a journey marked not only by significant successes but also by setbacks and disappointments. Along the way, he pays tribute to many colleagues, including several as renowned as himself.
Set against the backdrop of major political events, the narrative gains depth, vibrancy, and artistry. I believe this story will captivate many readers, especially young researchers, inspiring a profound sense of connection to the scientific process. The article’s genre does not permit comments or corrections, nor does it require them. I extend my gratitude to the author for providing such an engaging and enriching read.
Author Response
I would like to thank the Reviewer for their most favorable comments.
Reviewer 3 Report
Comments and Suggestions for Authors
I appreciate the privilege of reading this article before publication. I have enjoyed the whole story very much.
Author Response

(The authors gave the same response as above.)
